# Modified Albumin-Bilirubin Grade and Alpha-Fetoprotein Score (mALF Score) for Predicting the Prognosis of Hepatocellular Carcinoma after Hepatectomy

**DOI:** 10.3390/cancers14215292

**Published:** 2022-10-27

**Authors:** Masaki Kaibori, Kengo Yoshii, Kosuke Matsui, Hideyuki Matsushima, Hisashi Kosaka, Hidekazu Yamamoto, Kazunori Aoi, Takashi Yamaguchi, Katsunori Yoshida, Takeshi Hatanaka, Atsushi Hiraoka, Toshifumi Tada, Takashi Kumada, Mitsugu Sekimoto

**Affiliations:** 1Department of Surgery, Kansai Medical University, Osaka 573-1191, Japan; 2Department of Mathematics and Statistics in Medical Sciences, Kyoto Prefectural University of Medicine, Kyoto 602-8566, Japan; 3Department of Gastroenterology and Hepatology, Kansai Medical University, Osaka 573-1191, Japan; 4Department of Gastroenterology, Gunma Saiseikai Maebashi Hospital, Maebashi 371-0821, Japan; 5Gastroenterology Center, Ehime Prefectural Central Hospital, Matsuyama 790-0024, Japan; 6Department of Internal Medicine, Japanese Red Cross Himeji Hospital, Himeji 670-8540, Japan; 7Department of Nursing, Gifu Kyoritsu University, Ogaki 503-8550, Japan

**Keywords:** hepatocellular carcinoma, modified albumin-bilirubin grade and α-fetoprotein score, modified albumin-bilirubin grade, α-fetoprotein, prognosis, complication

## Abstract

**Simple Summary:**

Nutritional and oncological assessments are important for predicting prognosis in cancer. We developed a modified albumin-bilirubin grade and α-fetoprotein (mALF) score based on the modified albumin-bilirubin (mALBI) grade and α-fetoprotein (AFP) level. Our results indicate that the mALF score has better predictive value for prognosis and shows greater sensitivity for predicting risk of postoperative complications as compared with mALBI or AFP in patients undergoing hepatectomy for hepatocellular carcinoma.

**Abstract:**

We developed and evaluated a modified albumin-bilirubin grade and α-fetoprotein (mALF) score, a nutritional and oncological assessment tool for patients with hepatocellular carcinoma (HCC) after surgical resection. Patients (*n* = 480) who underwent R0 resection between 2010 and 2020 were analyzed retrospectively. The mALF score assigned one point for a modified albumin-bilirubin (mALBI) grade 2b or 3 and one point for an α-fetoprotein (AFP) level ≥ 100 ng/mL. Patients were classified by mALF scores of 0 (mALBI grade 1/2a, AFP < 100 ng/mL), 1 (mALBI grade 2b/3 or AFP ≥ 100 ng/mL), or 2 (mALBI grade 2b/3, AFP ≥ 100 ng/mL) points. Liver reserve deteriorated and cancer progressed with increasing score. Postoperative complications (Clavien–Dindo classification ≥ 3) differed significantly among groups. The 5-year recurrence-free survival (RFS) rates were 34.8%, 11.2%, and 0.0% for 0, 1, and 2 points, respectively (1 or 2 versus 0 points, *p* < 0.001). The 5-year overall survival (OS) rates were 66.0%, 29.7%, and 17.8% for 0, 1, and 2 points, respectively (1 or 2 versus 0 points, *p* < 0.001). The mALF score was an independent prognostic predictor of RFS and OS. In HCC, the mALF score was effective for predicting postoperative complications and long-term survival.

## 1. Introduction

Hepatocellular carcinoma (HCC) is the most common primary malignancy of the liver and the fifth most common of all malignancies [1]. In surgical treatment against malignant tumors, nutritional assessment is important for predicting prognosis. The prognostic nutritional index (PNI) [2,3], neutrophil/lymphocyte ratio (NLR) [4], platelet/lymphocyte ratio (PLR) [5], and controlling nutritional status (CONUT) score [6] were proposed as nutritional prognostic assessment tools. In addition, the Glasgow prognostic score (GPS) [7,8,9], defined based on C-reactive protein (CRP; 1.0 mg/dL) and serum albumin (3.5 g/dL) levels, was shown to be an important and useful nutritional assessment tool for predicting prognosis in patients with malignant tumors. Moreover, patients with a GPS score ≥ 1 have been reported to have a higher rate of complications following surgical resection for advanced gastric cancer compared with patients with a low GPS score (<1) [10]. Previously, we developed an improved GPS scoring method (neo-GPS) [11], based on the albumin-bilirubin (ALBI) grade, and reported that it identified the approximate borderline of amino-acid imbalance [12] and could be used instead of serum albumin for predicting prognosis following surgical resection. 

In contrast, many studies have found that α-fetoprotein (AFP) and protein induced by vitamin K absence or antagonist-II (PIVKA-II), which are tumor markers for HCC, predict postoperative prognosis [13,14,15]. It is important to establish a simple scoring system that reflects preserved liver function and oncologic prognostic factors. Therefore, the aim of the current study was to develop a new, simple score using both the modified ALBI grade, which indicates the approximate borderline of amino-acid imbalance, and the AFP level to predict prognosis following surgical resection.

## 2. Materials and Methods

### 2.1. Patients

The records of all patients with HCC who underwent liver resection between January 2010 and September 2020 at Kansai Medical University Hospital (Osaka, Japan) were screened. A total of 480 patients who underwent an R0 resection, defined as macroscopic removal of all tumors, were enrolled in this study. Among them, 429 were classified as Child–Pugh class A. A single surgeon who had performed more than 1500 hepatic resection procedures was responsible for all patients analyzed in this study. The study protocol was approved by the institutional ethics committee of Kansai Medical University (reference number: KMU 2021311).

### 2.2. Underlying Liver Disease

HCC was considered due to hepatitis C virus (HCV) infection in cases with positive anti-HCV findings, while that due to hepatitis B virus (HBV) infection was determined in cases positive for the HBV surface antigen. Underlying liver disease was judged as related to alcohol for patients with a history of alcohol abuse (≥60 g/day) [16].

### 2.3. Liver Function and Nutritional Status Assessments

Child–Pugh score/classification [17], ALBI grade [18,19], and modified albumin-bilirubin (mALBI) grade [20] were used for hepatic reserve function assessment. 

### 2.4. Clinicopathologic Variables, Treatment Algorithm for HCC, and Surgical Procedures

Patients were measured for the indocyanine green retention rate at 15 min (ICG-R15) and underwent conventional liver function tests prior to surgery. Patients also underwent measurement of levels of AFP and PIVKA-II. We used the updated treatment algorithm for HCC, which included a combination of five factors: liver function reserve, extrahepatic metastasis, vascular invasion, tumor number, and tumor size [21]. The degree of liver damage (including the ICG-R15) was used to determine the indication for hepatectomy. We summarized the new treatment algorithm as follows: patients with HCC with Child–Pugh class A/B liver function without extrahepatic metastasis or vascular invasion are recommended to receive one of three treatment regimens. First, either surgical resection or radiofrequency ablation is recommended with no priority for up to three tumors measuring ≤3 cm, or surgical resection is recommended as first-line therapy for a solitary tumor, regardless of size. Second, for up to three tumors measuring >3 cm, surgical resection is recommended as first-line therapy, and transarterial chemoembolization is recommended as second-line therapy. Third, for patients with HCC accompanied by vascular invasion without extrahepatic metastasis, a combination of embolization, hepatectomy, hepatic arterial infusion chemotherapy, and molecular targeted therapy is recommended. Treatment is selected for each patient according to the individual situation, including consideration of the following factors: liver function, the number and size of HCC lesions, and the extent of vascular invasion. 

The Brisbane terminology proposed by Strasberg et al. was used to classify surgical procedures [22]. Anatomic resection was defined as resection of the tumor together with the related portal vein branches and corresponding hepatic territory. Anatomic resection was classified as hemihepatectomy (resection of half of the liver), extended hemihepatectomy (hemihepatectomy plus removal of additional contiguous segments), sectionectomy (resection of two Couinaud subsegments [23]), or segmentectomy (resection of one Couinaud subsegment). All other non-anatomic procedures were classified as limited resections. Limited resection was used to manage both peripheral and central tumors. Because partial hepatectomy allows adequate surgical margins, it was used to manage peripheral tumors and those with extrahepatic growth. Conversely, because of the difficulty and risks associated with achieving adequate margins, enucleation was used to manage central tumors near the hepatic hilum or major vessels. Each specimen was reviewed by a senior pathologist who performed a histological review to confirm the final diagnosis.

### 2.5. Modified ALBI Grade and AFP Score (mALF Score)

We developed a simple score, the modified albumin-bilirubin grade and AFP score (mALF score). We assigned 1 point for an mALBI grade 2b or 3 and 1 point for a baseline AFP level ≥ 100 ng/mL. Accordingly, a patient was classified by an mALF score of 0 points (mALBI grade 1 or 2a and AFP < 100 ng/mL), 1 point (either mALBI grade 2b or 3 and AFP ≥ 100 ng/mL), and 2 points (both mALBI grade 2b or 3 and AFP ≥ 100 ng/mL).

### 2.6. Evaluation of Complications Following Surgical Resection

For the evaluation of complications associated with surgical resection, the Clavien–Dindo classification [24] was used, with grade ≥3 considered to be a significant complication.

### 2.7. Statistical Analysis

Continuous variables were classified into two categories using the median value. Three groups of clinical characteristics were compared using the chi-square test or Fisher’s exact test, as appropriate. The probabilities of recurrence-free survival (RFS) and overall survival (OS) after hepatectomy were calculated using the Kaplan–Meier method. Hazard ratios for RFS and OS, and their 95% confidence intervals (CIs) were estimated using univariate Cox analysis. Multivariate analysis was performed using Cox proportional hazards analysis. For all analyses, *p* values less than 0.05 were considered to indicate statistical significance. The discriminating ability of the scoring models was assessed for prognostic ability using Harrell’s concordance-index (c-index). The R version 4.1.2 (R Foundation for Statistical Computing, Vienna, Austria) was used to perform the statistical analyses. Survival analysis was executed with the R package “survival”.

## 3. Results

### 3.1. Comparison of Perioperative Characteristics in Three Groups Classified by the mALF Scoring System

The characteristics of the patients classified into three groups by mALF score are shown in Table 1. There were significant differences among the three groups in the following: platelets, albumin, prothrombin time, CRP, AFP, PIVKA-II, ICGR15, ALBI score, Fib4-index, Child–Pugh score, fibrosis stage, degree of differentiation, number of tumors, portal vein invasion, hepatic vein invasion, and mALBI. As the score increased, liver reserve function deteriorated, and liver cancer progressed.

Perioperative characteristics are shown in Table 2. The operative time and postoperative complications (Clavien–Dindo classification ≥3, postoperative hospital stay, and readmission within 30 days) differed significantly among the three groups.

### 3.2. Long-Term Survival

Patient survival was determined in the three groups, and the Kaplan–Meier curves demonstrated that the mALF score enabled satisfactory risk evaluations of survival (Figure 1). The 1-, 3-, and 5-year RFS rates of the groups were significantly different, at 78.3%, 50.2%, and 33.5%, respectively, for 0 points; at 60.5%, 31.4%, and 22.5%, respectively, for 1 point; and at 34.8%, 11.2%, and 0.0%, respectively, for 2 points (1 or 2 versus 0 points: *p* < 0.001; Figure 1A). The patients were divided into three risk classifications of OS, and the 1-, 3-, and 5-year OS rates of the groups were significantly different at 93.2%, 77.6%, and 67.4%, respectively, for 0 points; at 84.7%, 61.6%, and 43.0%, respectively, for 1 point; and at 66.0%, 29.7% and 17.8%, respectively, for 2 points (1 or 2 versus 0 points: *p* < 0.001; Figure 1B).

For comparison to our preoperative staging system, Kaplan–Meier curves for RFS and OS based on serum AFP levels and mALBI grade are presented in Figure 1C–F. The 1-, 3-, and 5-year RFS rates of the groups were significantly different, at 75.8%, 45.9%, and 30.1%, respectively, for AFP < 100 ng/mL and 49.9%, 28.1%, and 21.4%, respectively for AFP ≥ 100 ng/mL (<100 versus ≥100 groups: *p* < 0.001; Figure 1C), and the 1-, 3-, and 5-year OS rates of the groups were significantly different, at 91.7%, 75.1%, and 62.6%, respectively, for AFP < 100 mg/mL and at 78.8%, 50.9%, and 37.2%, respectively, for AFP ≥ 100 ng/mL (<100 versus ≥100 groups: *p* < 0.001; Figure 1D). The 1-, 3-, and 5-year RFS rates of the groups were significantly different, at 73.2%, 46.5%, and 32.1%, respectively, for mALBI 1 or 2A and at 56.2%, 22.5%, and 10.1%, respectively, for mALBI 2B or 3 (1 or 2A versus 2B or 3 groups: *p* < 0.001; Figure 1E), and the 1-, 3-, and 5-year OS rates of the groups were significantly different at 91.1%, 73.4%, and 62.2%, respectively, for mALBI 1 or 2A and at 79.4%, 53.4%, and 33.5%, respectively, for mALBI 2B or 3 (1 or 2A versus 2B or 3 groups: *p* < 0.001; Figure 1F).

In the comparison of RFS, the c-index was higher (0.592 versus 0.570 or 0.550) according to the mALF scoring system as compared with AFP or the mALBI grading system. Additionally, in a comparison of OS, the c-index was higher (0.621 versus 0.589 or 0.569) based on the mALF scoring system.

### 3.3. Univariate and Multivariate Analysis of Prognostic Factors for Long-Term Survival

Cox proportional hazards analysis revealed five independent prognostic predictors for both RFS and OS (Table 3): prothrombin time ≥ 87% (RFS: hazard ratio, 0.69; 95% CI, 0.52–0.90; *p* = 0.007, OS: hazard ratio, 0.68; 95% CI, 0.48–0.97; *p* = 0.034), PIVKA-II ≥ 107 mAU/mL (RFS: hazard ratio, 1.48; 95% CI, 1.12–1.94; *p* = 0.005, OS: hazard ratio, 1.67; 95% CI, 1.16–2.42; *p* = 0.006), number of tumors ≥ 2 (RFS: hazard ratio, 1.66; 95% CI, 1.24–2.21; *p* < 0.001, OS: hazard ratio, 1.46; 95% CI, 1.01–2.11; *p* = 0.047), mALF point 1 (RFS: hazard ratio, 1.35; 95% CI, 1.02–1.78; *p* = 0.036, OS: hazard ratio, 1.95; 95% CI, 1.36–2.80; *p* < 0.001), and mALF score 2 points (RFS: hazard ratio, 3.02; 95% CI, 1.83–4.98; *p* < 0.001, OS: hazard ratio, 3.67; 95% CI, 2.03–6.63; *p* < 0.001).

## 4. Discussion

Inflammatory biomarkers have a strong prognostic value in surgically treated patients with HCC; however, the underlying pathogenic mechanisms are not completely clear. Conversely, nutritional biomarkers predict the outcomes after hepatic resection for HCC but not after liver transplantation (LT). Pravisani et al. reported that at 1 year after LT, the nutritional status of patients with HCC who received LT significantly improved, although the inflammatory state tended to persist. HCC in LT candidates with high PLR and/or NLR may have a more aggressive biology, requiring these patients to be thoroughly and more carefully assessed in the pre-LT workup. Moreover, NLR and PLR may be also used as reliable prognostic parameters for long-term clinical surveillance [25,26].

The major findings of the present study are that mALBI grade 2b or 3 and AFP ≥ 100 ng/mL were identified as independent unfavorable prognostic factors in multivariate analysis and that we developed an easy and widely applicable score, the mALF score, for patients with HCC undergoing hepatic resection. The mALF score used a newly developed assessment tool based on hepatic function and a tumor factor. This mALF score showed the best c-index for predicting prognosis for OS and RFS among other AFP as a tumor factor and mALBI as an hepatic function index. Finally, the mALF score was prognostic for RFS and OS in multivariate analysis, and it was found to be a better predictor of high Clavien–Dindo classification (≥3; Table 2). Thus, it is thought that the mALF score can be an important prognostic predictive assessment tool in patients with HCC.

The liver damage system, which is treated as a more suitable assessment tool than the Child–Pugh classification because it uses ICG-R15, has been used generally for patients treated with surgical resection. In contrast, the ALBI score has been shown not only to have good relationships with ICG-R15 (r = 0.563, 95% CI: 0.550–0.570, *p* < 0.0001) [20] but also to behave as a more detailed assessment tool than the liver damage system [27]. According to data from a Japanese nationwide survey, the mALBI grade can evaluate preserved liver function more precisely and accurately than the Child–Pugh classification [20]. In addition, the mALBI grade can predict and stratify the prognosis of patients with HCC [20]. 

AFP is a well-known tumor marker and is widely used in the management of HCC, such as in surveillance, diagnosis, monitoring of treatment response, and prognosis [28]. An elevated AFP level shows a poor prognosis across all stages of HCC [29]. An elevated AFP level was also related to a high risk of tumor recurrence after surgical resection [13,14,15] and liver transplantation [30]. An analysis of transcriptome data, whole-exome sequencing data, and DNA methylome profiling demonstrated that AFP-high tumors showed a different phenotype characterized by poor differentiation, enrichment of progenitor features and enhanced proliferation compared with AFP-low tumors [31]. From the above findings, it is easy to understand that the mALF score, which uses the mALBI grade and AFP level, showed the best predictive values not only for prognosis, and but also for high Clavien–Dindo classification (≥3). Because a high rate of complications is expected when hepatectomy is planned in patients with HCC with high mALF (2 points), nutritional intervention should be considered in such patients. Kaido and colleagues reported the importance of nutritional intervention in liver transplantation [32]. Recently, muscle volume decline has been reported to be an important prognostic factor for recurrence and OS [33]. As hepatic reserve function worsened, the frequency of muscle abnormalities increased [34]. However, a relationship between mALF score and muscle abnormality was not elucidated in the present study because there was no information on muscle volume (pre-sarcopenia) in the present database. This study was limited by its retrospective nature. From now on, in addition to assessment using the mALF score, an assessment for sarcopenia should be performed, and further examination must be set to elucidate their relationship in the near future.

Patients with compensated cirrhosis and large liver functional reserve can always receive the most radical treatment in terms of potential survival benefit according to their tumor stage, provided that there is no significant extrahepatic disease. Conversely, a more detailed and individualized assessment should be carried out in patients with poorer liver functional reserve. Such an assessment must balance the expected antitumor efficacy of any given locoregional therapy with the risk of deterioration in liver function, particularly considering the risk of progression to frank Child–Pugh class B or borderline Child–Pugh class A; this may limit the possibility of receiving multiple lines of systemic therapy, for which the clinical potential is rapidly increasing. The most balanced and potentially most complete treatment strategy should be foreseen and designed from the outset, considering the potential for several lines of therapy and not just the most readily available treatment [35,36].

## 5. Conclusions

The newly developed mALF score, based on the mALBI grade and AFP level, is a valuable prognostic nutritional and oncological assessment tool for prediction of postoperative complications and prognosis in patients with HCC after surgical resection.

## Figures and Tables

**Figure 1 cancers-14-05292-f001:**
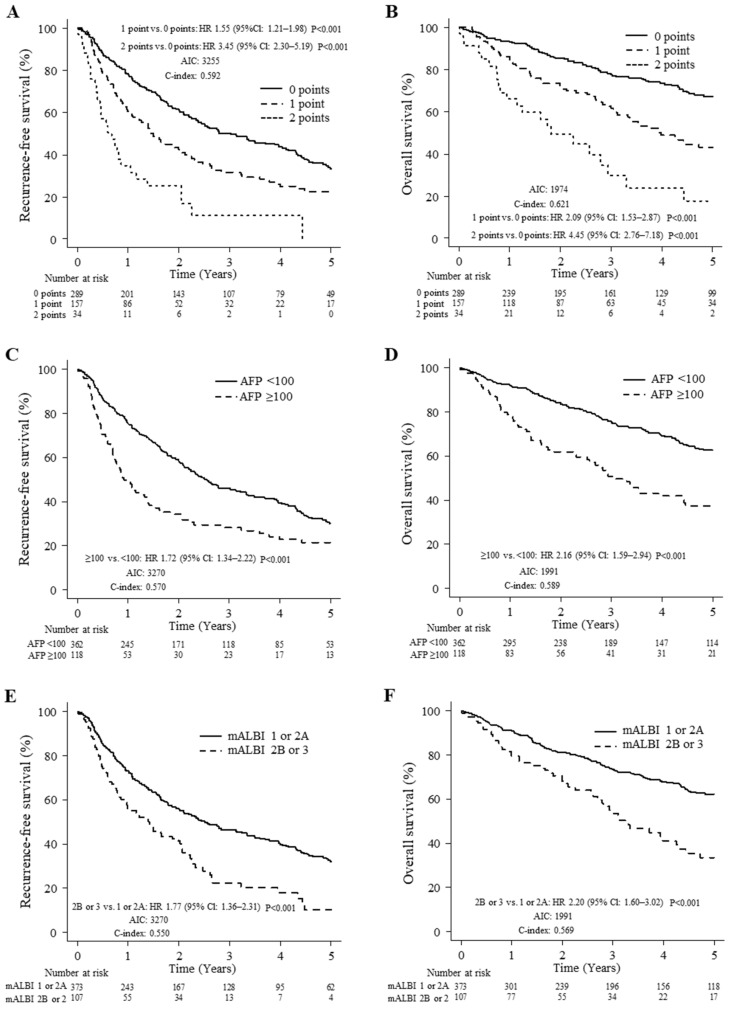
Comparison of mALF, AFP, and mALBI for prediction of recurrence-free survival (RFS) and overall survival (OS). (**A**) RFS in patients with mALF scores of 0, 1, and 2 points. (**B**) OS in patients with mALF scores of 0, 1, and 2 points. (**C**) RFS in patients with AFP <100 and ≥100 ng/mL. (**D**) OS in patients with AFP <100 and ≥100 ng/mL. (**E**) RFS in patients with mALBI 1 or 2A and 2B or 3. (**F**) OS in patients with mALBI 1 or 2A and 2B or 3.

**Table 1 cancers-14-05292-t001:** Clinical characteristics of 480 patients with hepatocellular carcinoma classified by mALF score.

Variable	0 Points (*n* = 289)		1 Point (*n* = 157)		2 Points (*n* = 34)		*p*	Variable	0 Points (*n* = 289)		1 Point (*n* = 157)		2 Points (*n* = 34)		*p*
Age, Years							0.423	PIVKA-II (mAU/mL)							0.004
<70	97	(34%)	58	(37%)	15	(44%)		<107	151	(54%)	74	(47%)	8	(24%)	
≥70	192	(66%)	99	(63%)	19	(56%)		≥107	127	(46%)	82	(53%)	25	(76%)	
Gender							0.224	ICGR15 (%)							0.017
Male	227	(79%)	112	(71%)	25	(74%)		<14.8	157	(55%)	65	(42%)	14	(41%)	
Female	62	(21%)	45	(29%)	9	(26%)		≥14.8	129	(45%)	91	(58%)	20	(59%)	
BMI, kg/m^2^							0.685	ALBI score							<0.001
<24	161	(56%)	94	(60%)	20	(59%)		Grade 1	206	(71%)	46	(29%)	0	(0%)	
≥24	128	(44%)	63	(40%)	14	(41%)		Grade 2	83	(29%)	106	(68%)	30	(88%)	
Alcohol							0.481	Grade 3	0	(0%)	5	(3%)	4	(12%)	
None	189	(65%)	111	(71%)	24	(71%)		Fib4-index							0.001
Positive	100	(35%)	46	(29%)	10	(29%)		Low	22	(8%)	4	(3%)	4	(12%)	
HBsAg							0.976	Middle	112	(39%)	39	(25%)	9	(26%)	
Negative	251	(87%)	135	(86%)	30	(88%)		High	155	(54%)	113	(72%)	21	(62%)	
Positive	38	(13%)	22	(14%)	4	(12%)		Child–Pugh score							<0.001
HCV Ab							0.375	<6	248	(86%)	75	(48%)	1	(3%)	
Negative	186	(65%)	92	(59%)	19	(56%)		≥6	41	(14%)	82	(52%)	33	(97%)	
Positive	102	(35%)	64	(41%)	15	(44%)		Fibrosis stage							0.002
Platelet (×10^4^/μL)							0.007	f0 or 1 or 2 or 3	206	(72%)	87	(57%)	25	(81%)	
<15.5	130	(45%)	94	(60%)	14	(41%)		f4	80	(28%)	65	(43%)	6	(19%)	
≥15.5	159	(55%)	63	(40%)	20	(59%)		Degree of differentiation							0.043
Albumin, g/dL							<0.001	Well or moderate	8	(3%)	6	(4%)	4	(13%)	
<3.9	70	(24%)	103	(66%)	34	(100%)		Poor	263	(97%)	139	(96%)	27	(87%)	
≥3.9	219	(76%)	54	(34%)	0	(0%)		Tumor size, cm							0.292
Total bilirubin, mg/dL							0.372	<3.0	116	(40%)	59	(38%)	9	(26%)	
<0.8	149	(52%)	73	(46%)	14	(41%)		≥3.0	173	(60%)	98	(62%)	25	(74%)	
≥0.8	140	(48%)	84	(54%)	20	(59%)		Number of tumors							<0.001
ALT, IU/L							0.355	Solitary	243	(84%)	111	(71%)	22	(65%)	
<29	147	(51%)	75	(48%)	13	(38%)		Multiple	46	(16%)	46	(29%)	12	(35%)	
≥29	142	(49%)	82	(52%)	21	(62%)		Portal vein invasion							0.028
Prothrombin time, %							0.002	Negative	119	(42%)	52	(35%)	7	(21%)	
<87	123	(43%)	89	(57%)	23	(68%)		Positive	163	(58%)	98	(65%)	27	(79%)	
≥87	165	(57%)	68	(43%)	11	(32%)		Hepatic vein invasion							0.031
CRP, mg/dL							<0.001	Negative	196	(70%)	89	(61%)	17	(50%)	
<0.1	162	(56%)	69	(44%)	5	(15%)		Positive	86	(30%)	57	(39%)	17	(50%)	
≥0.1	127	(44%)	88	(56%)	29	(85%)		mALBI							<0.001
AFP, ng/mL							<0.001	1 or 2a	289	(100%)	84	(54%)	0	(0%)	
<100	289	(100%)	73	(46%)	0	(0%)		2b or 3	0	(0%)	73	(46%)	34	(100%)	
≥100	0	(0%)	84	(54%)	34	(100%)									

Data are shown as *n* (%). mALF: modified albumin-bilirubin grade and α-fetoprotein; BMI: body mass index; HBsAg: hepatitis B surface antigen; HCV Ab: hepatitis C virus antibody; ALT: alanine aminotransferase; CRP: C-reactive protein; AFP: α-fetoprotein; PIVKA-II: protein induced by vitamin K absence or antagonist-II; ICGR15: indocyanine green retention rate at 15 min; ALBI: integration of albumin-bilirubin; mALBI: modified integration of albumin-bilirubin.

**Table 2 cancers-14-05292-t002:** Perioperative characteristics of 480 patients with hepatocellular carcinoma classified by mALF score.

Variable	0 Points (*n* = 289)		1 Point (*n* = 157)		2 Points (*n* = 34)		*p*
Surgical Procedure							0.077
Non-anatomic or segmentectomy	141	(49%)	79	(50%)	10	(29%)	
Sectionectomy or more than hemihepatectomy	148	(51%)	78	(50%)	24	(71%)	
Open, Laparoscopic							0.344
Open	206	(71%)	110	(70%)	28	(82%)	
Laparoscopic	83	(29%)	47	(30%)	6	(18%)	
Operative time, min							0.016
<330	151	(52%)	81	(52%)	9	(26%)	
≥330	138	(48%)	76	(48%)	25	(74%)	
Operative blood loss, ml							0.090
<617	151	(52%)	78	(50%)	11	(32%)	
≥617	138	(48%)	79	(50%)	23	(68%)	
Postoperative complications (Clavien–Dindo classification ≥3)							<0.001
No	249	(86%)	111	(71%)	21	(62%)	
Yes	40	(14%)	46	(29%)	13	(38%)	
Postoperative hospital stay, days							<0.001
<14	153	(53%)	67	(43%)	5	(15%)	
≥14	136	(47%)	90	(57%)	29	(85%)	
Readmission within 30 days							0.008
No	272	(95%)	144	(92%)	24	(80%)	
Yes	13	(5%)	13	(8%)	6	(20%)	

mALF: modified albumin-bilirubin grade and alpha-fetoprotein.

**Table 3 cancers-14-05292-t003:** Cox proportional hazards regression analysis for recurrence-free survival and overall survival in patients with hepatocellular carcinoma who underwent hepatic resection.

Variable	Univariate Analysis			Multivariate Analysis			Univariate Analysis			Multivariate Analysis		
	HR	(95% CI)	*p*	HR	(95% CI)	*p*	HR	(95% CI)	*p*	HR	(95% CI)	*p*
Age ≥ 70 years (versus < 70 years)	0.99	(0.78–1.25)	0.918	0.95	(0.71–1.25)	0.695	1.18	(0.87–1.60)	0.287	1.09	(0.75–1.56)	0.655
HCV Ab Positive (versus Negative)	1.05	(0.83–1.33)	0.669	0.96	(0.73–1.28)	0.796	1.10	(0.81–1.48)	0.536	0.84	(0.58–1.22)	0.363
Platelets ≥ 15.5 (versus <15.5 × 10^4^/μL)	1.13	(0.90–1.42)	0.295	1.27	(0.92–1.77)	0.146	1.14	(0.85–1.53)	0.364	1.39	(0.92–2.12)	0.122
Albumin ≥ 3.9 (versus <3.9 g/dL)	0.54	(0.43–0.69)	<0.001				0.42	(0.31–0.56)	<0.001			
ALT ≥ 29 (versus <29 IU/L)	1.09	(0.87–1.37)	0.461	1.03	(0.79–1.35)	0.806	1.00	(0.74–1.33)	0.973	0.97	(0.69–1.38)	0.884
Prothrombin time ≥ 87 (versus < 87 %)	0.66	(0.53–0.83)	<0.001	0.69	(0.52–0.90)	0.007	0.63	(0.47–0.84)	0.002	0.68	(0.48–0.97)	0.034
AFP ≥ 100 (versus < 100 ng/mL)	1.72	(1.34–2.22)	<0.001				2.16	(1.59–2.94)	<0.001			
PIVKA-II ≥ 107 (versus < 107 mAU/mL)	1.68	(1.33–2.12)	<0.001	1.48	(1.12–1.94)	0.005	2.04	(1.51–2.77)	<0.001	1.67	(1.16–2.42)	0.006
ICGR15 ≥ 14.8 (versus < 14.8%)	1.14	(0.91–1.44)	0.253	0.87	(0.65–1.17)	0.361	1.40	(1.04–1.88)	0.025	1.16	(0.80–1.69)	0.430
ALBI score Grade 2 (versus 1)	1.88	(1.49–2.37)	<0.001				2.61	(1.92–3.55)	<0.001			
ALBI score Grade 3 (versus 1)	2.59	(1.14–5.90)	0.023				3.24	(1.18–8.94)	0.023			
Fib4-index Middle (versus Low)	1.12	(0.67–1.87)	0.675	1.21	(0.68–2.15)	0.524	1.46	(0.70–3.06)	0.317	1.55	(0.68–3.50)	0.296
Fib4-index High (versus Low)	1.35	(0.82–2.22)	0.235	1.58	(0.84–2.98)	0.154	1.86	(0.91–3.81)	0.089	2.42	(0.99–5.93)	0.053
Fibrosis stage f4 (versus f0 or 1 or 2 or 3)	1.12	(0.88–1.43)	0.358	1.19	(0.85–1.65)	0.309	1.00	(0.73–1.38)	0.977	1.12	(0.73–1.72)	0.601
Degree of differentiation Poor (versus Well or Mode)	0.89	(0.48–1.68)	0.730	0.93	(0.46–1.85)	0.826	0.80	(0.35–1.81)	0.590	0.63	(0.25–1.54)	0.309
Tumor size ≥3.0 (versus <3.0 cm)	1.38	(1.09–1.75)	0.008	1.21	(0.90–1.62)	0.216	1.44	(1.05–1.97)	0.022	1.04	(0.71–1.53)	0.847
Number of tumors ≥ 2 (versus 1)	1.90	(1.48–2.44)	<0.001	1.66	(1.24–2.21)	<0.001	1.75	(1.28–2.39)	<0.001	1.46	(1.01–2.11)	0.047
Portal vein invasion Positive (versus Negative)	1.42	(1.11–1.81)	0.006	1.21	(0.91–1.61)	0.193	1.77	(1.26–2.48)	0.001	1.52	(1.01–2.28)	0.043
mALBI 2b or 3 (versus 1 or 2a)	1.77	(1.36–2.31)	<0.001				2.20	(1.60–3.02)	<0.001			
Surgical procedure Sectionectomy or more than hemihepatectomy (versus non-anatomic or segmentectomy)	1.32	(1.05–1.67)	0.016	0.96	(0.72–1.28)	0.800	1.50	(1.12–2.02)	0.007	1.33	(0.91–1.93)	0.136
Operative blood loss ≥ 617 (versus <617 mL)	1.35	(1.08–1.70)	0.009	1.19	(0.91–1.55)	0.195	1.51	(1.12–2.03)	0.007	1.16	(0.82–1.64)	0.398
mALF 1 point (versus 0 points)	1.55	(1.21–1.98)	<0.001	1.35	(1.02–1.78)	0.036	2.09	(1.53–2.86)	<0.001	1.95	(1.36–2.80)	<0.001
mALF 2 points (versus 0 points)	3.45	(2.30–5.19)	<0.001	3.02	(1.83–4.98)	<0.001	4.45	(2.76–7.18)	<0.001	3.67	(2.03–6.63)	<0.001

HR; hazard ratio; CI; confidence interval; HCV Ab: hepatitis C virus antibody; ALT: alanine aminotransferase; AFP: alpha-fetoprotein; PIVKA-II: protein induced by vitamin K absence or antagonist-II; ICGR15: indocyanine green retention rate at 15 min; ALBI: integration of albumin-bilirubin; mALBI: modified integration of albumin-bilirubin; mALF: modified albumin-bilirubin grade and α-fetoprotein.

## Data Availability

Due to the nature of this research, participants in this study could not be contacted regarding whether the findings could be shared publicly; thus, supporting data are not available. The datasets generated and/or analyzed for the current study are not publicly available due to the nature of the research, as noted above.

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
