# Peer review of "Modified Albumin-Bilirubin Grade and Alpha-Fetoprotein Score (mALF Score) for Predicting the Prognosis of Hepatocellular Carcinoma after Hepatectomy"

_cancers, 2022, doi:10.3390/cancers14215292_

Round 1

Reviewer 1 Report

In this study, the authors developed and evaluated the modified Albumin-biLirubin grade and alpha-Fetoprotein (mALF) score, in patients with hepatocellular carcinoma (HCC) after surgical resection. Overall, they studied 480 patients who underwent resection. The mALF score assigns 1 point for a modified albu33 min-bilirubin (mALBI) grade 2b or 3 and 1 point for an alpha-fetoprotein (AFP) level ≥100 ng/ml. Patients were classified by mALF score 0 points (mALBI grade 1/2a and AFP <100 ng/ml), 1 point (mALBI grade 2b/3 or AFP ≥100 ng/ml), and 2 points (mALBI grade 2b/3 and AFP ≥100 ng/ml). Liver reserve deteriorated and cancer progressed with increasing score. Postoperative complications (Clavien-Dindo classification ≥3) differed significantly among groups. The 5-year recurrence-free survival (RFS) rate was 34.8%, 11.2%, and 0.0% for 0, 1, and 2 points, respectively (1 or 2 vs. 0 points, 39 p<0.001). The 5-year overall survival (OS) rate was 66.0%, 29.7%, and 17.8% for 0, 1, and 2 points, 40 respectively (1 or 2 vs. 0 points, p<0.001). They concluded that mALF score is an independent prognostic predictor of RFS and OS and it is an effective tool for prediction of postoperative complications and long-term survival.

The study is of interest and of clinical impact as the new proposed tool is easy to use in clinical practice. I have only minor points to suggest.

The assessmente of liver functional reserve in HCC patients is a topic of major impact as it affects the treatment allocation. In this regard, the authors should recall previous attempts to better assess the liver function in HCC patients as well discussed in a comprehensive review (Non-transplant therapies for patients with hepatocellular carcinoma and Child-Pugh-Turcotte class B cirrhosis. Lancet Oncol. 2017 Feb;18(2):e101-e112), and more recently with a new proposed treatment allocation system (The importance of liver functional reserve in the non-surgical treatment of hepatocellular carcinoma. J Hepatol. 2022 May;76(5):1185-1198.).

Author Response

Responses to the comments from Reviewer #1

We thank the reviewer for the valuable comments.

Comment

The assessmente of liver functional reserve in HCC patients is a topic of major impact as it affects the treatment allocation. In this regard, the authors should recall previous attempts to better assess the liver function in HCC patients as well discussed in a comprehensive review (Non-transplant therapies for patients with hepatocellular carcinoma and Child-Pugh-Turcotte class B cirrhosis. Lancet Oncol. 2017 Feb;18(2):e101-e112), and more recently with a new proposed treatment allocation system (The importance of liver functional reserve in the non-surgical treatment of hepatocellular carcinoma. J Hepatol. 2022 May;76(5):1185-1198.).

Response

We appreciate this comment from the reviewer, and we agree with the reviewer’s point. We have added this to the Discussion section of the revised manuscript, as follows:

Discussion (lines 269–280)

Patients with compensated cirrhosis and large liver functional reserve can always receive the most radical treatment in terms of potential survival benefit according to their tumor stage, provided that there is no significant extrahepatic disease. Conversely, a more detailed and individualized assessment should be carried out in patients with poorer liver functional reserve. Such assessment must balance the expected antitumor efficacy of any given locoregional therapy with the risk of deterioration in liver function, particularly considering the risk of progression to frank Child-Pugh class B or borderline Child-Pugh class A; this may limit the possibility of receiving multiple lines of systemic therapy, for which the clinical potential is rapidly increasing. The most balanced and potentially most complete treatment strategy should be foreseen and designed from the outset, considering the potential for several lines of therapy and not just the most readily available treatment [35, 36].

New references

  1. Granito, A; Bolondi, L. Non-transplant therapies for patients with hepatocellular carcinoma and Child-Pugh-Turcotte class B cirrhosis. Lancet Oncol. 2017, 18(2), e101-e112.
  2. D'Avola, D.; Granito, A.; de la Torre-Aláez, M.; Piscaglia, F. The importance of liver functional reserve in the non-surgical treatment of hepatocellular carcinoma. Hepatol.2022, 76(5), 1185–1198. 

Reviewer 2 Report

The authors showed the a new score based on modified Albumin-biLirubin grade and alpha-Fetoprotein (mALF) is able to predict both RFS and OS after resection for HCC; it is already well-known that both AFP and mALBI are both predictors of outcome in HCC, however the authors demonstrated that adding together these two parameters the power of predicting RFS and OS is higher. Those results introduce a new composite scoring system in the already crowded scenario of HCC. The paper could be improved in the discussion section by highlighting much more the role of inflammatory and nutritional based scoring system in the scenario of HCC both for resection and transplantation since the latter remains probably the most powerful therapy for the treatment of HCC (see Cancers (Basel). 2021 Jan 29;13(3):513. doi: 10.3390/cancers13030513 and Clin Transplant. 2020 Mar;34(3):e13786. doi: 10.1111/ctr.13786. Epub 2020 Feb 3.)

Author Response

Responses to the comments from Reviewer #2

We thank the reviewer for the valuable comments.

Comments

The paper could be improved in the discussion section by highlighting much more the role of inflammatory and nutritional based scoring system in the scenario of HCC both for resection and transplantation since the latter remains probably the most powerful therapy for the treatment of HCC (see Cancers (Basel). 2021 Jan 29;13(3):513. doi: 10.3390/cancers13030513 and Clin Transplant. 2020 Mar;34(3):e13786. doi: 10.1111/ctr.13786. Epub 2020 Feb 3.)

Response

We appreciate this comment from the reviewer, and we agree with the reviewer’s suggestion. Accordingly, we have added the following text to the revised Discussion section:

Discussion (lines 219–227)

Inflammatory biomarkers have a strong prognostic value in surgically treated patients with HCC; however, the underlying pathogenic mechanisms are not completely clear. Conversely, nutritional biomarkers predict the outcomes after hepatic resection for HCC but not after liver transplantation (LT). Pravisani et al. reported that at 1 year after LT, the nutritional status of patients with HCC who received LT significantly improved, although the inflammatory state tended to persist. HCC in LT candidates with high PLR and/or NLR may have a more aggressive biology, requiring these patients to be thoroughly and more carefully assessed in the pre-LT workup. Moreover, NLR and PLR may be also used as reliable prognostic parameters for long-term clinical surveillance [25, 26].

New references

  1. Pravisani, R.; Mocchegiani, F.; Isola, M.; Lorenzin, D.; Adani, G.L.; Cherchi, V.; Righi, E.; Terrosu, G.; Vivarelli, M.; Risaliti, A.; et al. Controlling nutritional status score does not predict patients' overall survival or hepatocellular carcinoma recurrence after deceased donor liver transplantation. Transplant. 2020, 34(3), e13786.
  2. Pravisani, R.; Mocchegiani, F.; Isola, M.; Lorenzin, D.; Adani, G.L.; Cherchi, V.; De Martino, M.; Risaliti, A.; Lai, Q.; Vivarelli, M.; et al. Postoperative trends and prognostic values of inflammatory and nutritional biomarkers after liver transplantation for hepatocellular carcinoma. Cancers (Basel). 2021, 13(3), 513.
